# Human Skin as an Ex Vivo Model for Maintaining *Mycobacterium leprae* and Leprosy Studies

**DOI:** 10.3390/tropicalmed9060135

**Published:** 2024-06-20

**Authors:** Natália Aparecida de Paula, Marcel Nani Leite, Daniele Ferreira de Faria Bertoluci, Cleverson Teixeira Soares, Patrícia Sammarco Rosa, Marco Andrey Cipriani Frade

**Affiliations:** 1Department of Cell and Molecular Biology and Pathogenic Bioagents, Ribeirão Preto Medical School, University of São Paulo, Ribeirão Preto14049-900, Brazil; npbiomed@yahoo.com.br; 2Dermatology Division, Department of Medical Clinics, Ribeirão Preto Medical School, University of São Paulo, Ribeirão Preto 14049-900, Brazil; marcelnani@gmail.com; 3Reference Center for Sanitary Dermatology with Emphasis on Leprosy, Ribeirão Preto Medical School, University of São Paulo, Ribeirão Preto 14049-900, Brazil; 4Department of Anatomic Pathology, Lauro de Souza Lima Institute, Bauru 17034-971, Brazil; danibertoluci@hotmail.com (D.F.d.F.B.); clev.blv@terra.com.br (C.T.S.); 5Division of Research and Education, Lauro de Souza Lima Institute, Bauru 17034-971, Brazil; sammarcop@hotmail.com

**Keywords:** *Mycobacterium leprae*, leprosy, hOSEC, ex vivo skin culture, animal-use alternatives, tissue culture techniques

## Abstract

The in vitro cultivation of *M. leprae* has not been possible since it was described as causing leprosy, and the limitation of animal models for clinical aspects makes studies on leprosy and bacteria–human host interaction a challenge. Our aim was to standardize the ex vivo skin model (hOSEC) to maintenance and study of *M. leprae* as an alternative animal model. Bacillary suspensions were inoculated into human skin explants and sustained in DMEM medium for 60 days. Explants were evaluated by RT-PCR-16SrRNA and cytokine gene expression. The viability and infectivity of bacilli recovered from explants (D28 and D60) were evaluated using the Shepard’s model. All explants were RT-PCR-16SrRNA positive. The viability and infectivity of recovered bacilli from explants, analyzed after 5 months of inoculation in mice, showed an average positivity of 31%, with the highest positivity in the D28 groups (80%). Furthermore, our work showed different patterns in cytokine gene expression (TGF-β, IL-10, IL-8, and TNF-α) in the presence of alive or dead bacilli. Although changes can be made to improve future experiments, our results have demonstrated that it is possible to use the hOSEC to maintain *M. leprae* for 60 days, interacting with the host system, an important step in the development of experimental models for studies on the biology of the bacillus, its interactions, and drug susceptibility.

## 1. Introduction

*Mycobacterium leprae* (*M. leprae*), or Hansen bacillus, described by Gerhard Henrik Armauer Hansen in 1873 [1], was the first bacterial pathogen identified and considered to be the cause of a human infectious disease; however, the fact that *M. leprae* cannot be grown in vitro still poses a challenge to researchers. *M. leprae* subverts the immune defense of the host, infects macrophages and Schwann cells of the peripheral nerve system in the skin [2,3], and causes leprosy, a chronic dermatoneurological disease, which continues to carry a strong social stigma and is widespread throughout the world, mainly in Brazil and India [4].

Depending on the immunological response (humoral and cellular) of the host, leprosy presents a complex clinical spectrum [5]. After having contact with *M. leprae*, certain individuals do not become ill. Other individuals present with the disease but do not have severe symptoms; the treatment of these individuals is efficient and uncomplicated. Other individuals, however, become ill, present important symptoms and complications during and/or after treatment (mainly in late diagnosis), and a few may become resistant to drugs [6].

One of the causes of delayed diagnosis of leprosy and the difficulty in studying *M. leprae* and its interactions with the host is related to the inability of these mycobacteria to be cultured in vitro. Numerous attempts to cultivate in vitro using several different culture media have been unsuccessful in the maintenance of the live mycobacterium, in growth, and induction of the pathology [7,8,9].

Currently, the only method for maintaining viable *M. leprae* in the laboratory is by in vivo inoculation, mainly passages into athymic nude mice, a laborious and time-consuming technique [10].

Although this in vivo experimental model has allowed important advances in *M. leprae* studies, it has limitations because it does not thoroughly reproduce the disease. The impossibility of in vitro *M. leprae* culture has hampered the study of basic aspects involved in disease transmission, genetic and immunological factors involved in resistance/susceptibility to the disease, and the testing of new therapeutic targets.

Although we have a model for cultivating bacilli in mouse foot pads, it is important to highlight the current advances in policies and laws related to restricting the use of animals in scientific experiments, through alternative methods [11,12].

Thus, human organotypic skin explant culture (hOSEC) is an ex vivo model of human skin, an alternative method, applied to study the skin and some interventions. Here, we used hOSEC to standardize its use in the maintenance of *M. leprae*, since the skin is the natural habitat of the bacillus, and we showed that the use of this model is feasible to maintain the viability and pathogenicity of *M. leprae* to study the interaction between the bacillus and the human skin.

## 2. Materials and Methods

### 2.1. hOSEC (Human Organotypic Skin Explant Culture)

Fragments of healthy human skin were obtained from skin remaining from tummy tuck surgery after the informed consent of patients in accordance with the ethical guidelines of the Research Ethics Committee of the Clinics Hospital at Ribeirão Preto Medical School, University of São Paulo (protocol number 1.744.888/2016).

Soon after the surgery, the skin tissues were manipulated inside a laminar flow hood and placed in phosphate-buffered saline (PBS, pH 7.2) plus 1.5% antibiotic solution (100 U/mL penicillin and 100 mg/mL streptomycin; Gibco^®^, Grand Island, NY, USA) overnight at 4 °C for decontamination. Later, the subcutaneous tissue was removed with scissors and the full-thickness skin (epidermis and dermis) was cut with an 8 mm diameter biopsy punch. The explants were placed with the dermal side facing down on pieces of filter paper (80 g/m^2^, 26 l/s m^2^ air permeability, 25 μm porosity) supported by metal grids in six-well culture plates; each grid supported three explants, similar to as described by Frade and collaborators [13] (Figure 1). Approximately five milliliters of Dulbecco’s modified Eagle medium (DMEM; Gibco^®^, Grand Island, NY, USA) supplemented with 10% fetal bovine serum (FBS; Gibco^®^, Grand Island, NY, USA) and 1% antibiotic solution (100 U/mL penicillin and 100 mg/mL streptomycin) was added to each well until it reached the dermis. Every third day, 2 mL of exhausted medium was replaced with fresh medium. Experiments were performed on skin from four different individuals.

### 2.2. Inoculation of M. leprae

Before the explants were placed on the plates, each one was inoculated with 25 μL of a suspension containing 1.5 × 10^4^
*M. leprae* bacilli. The bacilli in the suspension were obtained from continuous passage of the *M. leprae* Thai-53 strain into the foot pads of athymic mice (NU-Foxn1nu) from the Lauro de Souza Lima Institute in Bauru, São Paulo-Brazil. The suspension was obtained following the method described by Trombone and collaborators [10]. Briefly, a mouse was euthanized, the foot pads were removed and cleaned with 70° ethanol, the bone tissue was discarded, and the remaining tissue was cut into small pieces and transferred to a tube containing Hanks’ solution. The tissue was homogenized, trypsin was added, and the tube was incubated at 37 °C for 1 h. After trypsin inactivation, the suspension was centrifuged, and the supernatant was discarded. The pellet was re-suspended in 1 mL of saline solution (0.9% NaCl) and homogenized with a syringe and 12 × 0.7 needle. The bacilli were counted on a glass slide after Ziehl–Neelsen staining (ZN) and the percentage of viability determined by the Live/Dead BacLight Bacterial Viability kit (ThermoFisher Scientific, Burlington, ON, Canada, cat: L7007). The suspensions used had at least 86% viability, and the dilution for inoculation was based only on the number of live bacilli determined by the Live/Dead BacLight kit (integral membrane). Equal volumes with equal amounts of bacilli were separated, one directed towards the inoculation of live bacilli and the other towards dead bacilli, which were obtained after autoclaving (autoclave at 121 °C for 40 min). An aliquot of the suspension was tested for contaminant microorganism growth on brain heart infusion (BHI) medium and Sabouraud medium, both solid and liquid, under aerobic and anaerobic conditions.

For inoculation into explants, the viable bacillary suspension (VML) was diluted in saline solution, and intradermal injection was performed using a microsyringe (Hamilton^®^ Company—Reno, NV, USA). Other explants were inoculated with the same number of bacilli previously inactivated by autoclaving (DML), and saline solution (25 µL) was inoculated into control explants. After inoculation, the explants were placed on plates as previously described and incubated at 37 °C in 5% CO_2_ for 4, 7, 14, 28, and 60 days. Three explants from each group were collected for analysis using three different protocols (histomorphology, viability, and enumeration/inoculation) starting at the initial time point (D0) and then on other days (D4, D7, D14, D28, and D60).

### 2.3. Histomorphology Analysis of hOSEC

The explants were embedded in paraffin for histomorphological analysis, and the slides were prepared with 4 μm sections and hematoxylin and eosin staining (HE). The analysis was performed with a Leica^®^ DM-4000B optical microscope with a Leica^®^ DFC 280 camera connected to a computer using Leica Application Suite (LAS^®^) for capturing images (Leica Microsystems, Mannheim, Germany).

### 2.4. Molecular Analysis of hOSEC

Explants were placed in TRIzol^®^ (Life Technologies, Carlsbad, CA, USA) and maintained at −80 °C for molecular biology analysis. To isolate the RNA, the explants were thawed and thoroughly macerated with 2 mL of TRIzol^®^ using a tissue homogenizer (Omni TH tissue homogenizer—Kennesaw, GA, USA), with the tubes maintained on ice during the process.

The RNA isolation technique followed the recommendations of the TRIzol^®^ reagent manufacturer; briefly, to every 1 mL of macerated sample was added 200 μL of cold chloroform (J. T Baker. cat.: 9180-02), followed by shaking and centrifugation (12,000× *g* at 4 °C for 15 min). The upper aqueous phase was transferred to a new microtube containing 500 μL of isopropanol, incubated at −80 °C for approximately 24 h, and centrifuged at 12,000× *g* at 4 °C for 10 min. The supernatant was discarded, and the pellet was washed with 1000 μL of 75% ethanol followed by centrifugation at 7500× *g* at 4 °C for 5 min. After drying, the pellet was eluted with 30 μL of diethyl pyrocarbonate water (DEPC) (Sigma-Aldrich, Saint Louis, MO, USA, cat.: D5758), and the amount of RNA was determined using a NanoVue^®^ Plus spectrophotometer (GE Healthcare Life Sciences, Buckinghamshire, UK).

DNAse (RQ1 RNase-Free DNase, Promega, Madison, WI, USA, cat.: M6101) was added to the RNA samples for DNA digestion according to the manufacturer’s protocol (Promega, cat.: M6101), and 500 ng were reverse-transcribed using random primers and the GoScript^®^ Reverse Transcriptase System (Promega, Madison, WI, USA, cat.: A5001) according to the manufacturer’s protocol and incubated in a Bio-Rad T100 thermal cycler (Bio-Rad Laboratories, Foster City, CA, USA, cat.: 1861096). The cDNA was diluted 1:2, and 5 μL was used for RT-qPCR.

PCR was performed on a CFX96^®^ Real-Time PCR Detection System (Bio-Rad, CFX96^®^ Touch System, Foster City, CA, USA, cat.: 184-5096) using 12.5 μL SYBR Green Master Mix (2x) (GoTaq qPCR Master Mix, Promega, Madison, WI, USA, cat.: A6002), 0.5 μL of each primer (10 μM), and 6.5 μL of nuclease-free water. The primer sequences and cycling protocol are shown in Table 1.

The cycle threshold (CT) value, positivity to sample, was considered only if the melting temperature (TM) was equal to the reaction control [18,19,20]. The expression rate of cytokines (TGF-β, TNF-α, IFN-γ, IL-1β, IL-10, and IL-8) was estimated for each explant with viable *M. leprae* and dead *M. leprae*, comparing with a saline group, using the 2^ΔΔCT^ formula (ΔΔCT = ΔCT test − ΔCT saline; ΔCT = target gene CT value − reference gene CT value) [19], and 18S rRNA was used as a reference gene, and the saline group to normalize.

### 2.5. Viability Using In Vivo Model

Following 28 and 60 days of hOSEC, the explants were processed to harvest bacilli for inoculation into the foot pads of athymic mice. The fragments were cut into smaller pieces using scissors, transferred to tubes containing 1000 μL of saline solution, and homogenized by three pulses of a tissue homogenizer (Tissue Homogenizer Omni TH^®,^ Kennesaw, GA, USA) at a speed of 4 (14,450 rpm) for 15 s. The tubes were always maintained on ice. Then, the homogenates were filtered through a cell strainer to eliminate the remaining debris, centrifuged at 12,000× *g* at 4 °C for 10 min, and suspended in 200 μL of saline.

These bacilli suspensions were taken, cooled, by road transport to the ILSL (Bauru, SP, Brazil), where, following Shepard model [21], two hind foot pads of mice were inoculated with 30 μL of the obtained bacillary suspension using a 30 G needle and insulin syringe. For each analysis time (D28 and D60), there were three fragments of each skin, and each fragment generated a 200 μL suspension that was divided into three animals (six foot pads/fragment). There was a minimum of three animals per fragment suspension (D28 and D60) and three fragments for each time, for each skin. After five months, the animals were euthanized, and their foot pads were removed for molecular and histological analysis, Fite Faraco (FF) and Ziehl–Neelsen. ZN was performed on one macerated foot pad, FF was performed on half of the second foot pad, and RT-PCR 16S rRNA was performed on the other half of the second foot pad.

The procedures were in accordance with the Ethical Principles in Animal Research and were approved by the Local Animal Ethical Committee of the Ribeirão Preto Medical School, University of São Paulo (protocol number 026/2015-1).

### 2.6. Statistical Analysis

Statistical analysis was performed using the GraphPad Prism 5 program, using one-way ANOVA followed by Tukey’s Test, for comparisons between all groups and T-test for comparisons between two groups, with a 95% confidence interval, and values of significant *p* were: * *p* < 0.05, ** *p* < 0.01, and *** *p* < 0.001.

## 3. Results

### 3.1. Characterization of Donor Skin and Histomorphology

The skin used for hOSEC was obtained from skin remaining from tummy tuck surgery in four white subjects (three females and one male) between the ages of 25 and 56 years (median 46.5 years), unrelated to known leprosy patients.

Upon histological examination, we did not observe any difference between skin with or without bacilli; in general, the hOSEC skin showed scattered cells and structures of glands, vessels, muscles, nerve fillets, and hair follicles, which were observed throughout the culture time. The skin maintained its natural architecture until the 7th day and was very similar to that observed at the initial time point, maintaining all layers of the epidermis, spinous, granulosa, and cornea without visible changes to the dermo–epidermal junction. At subsequent time points, there was a progressive decrease in the number of keratinocyte layers, the epidermis was more rectified, and the corneal layer thickness was increased; however, the dermo–epidermal junction and the basal layer were maintained. The papillary dermis was easily observed until the 14th day; from the 28th day on, it became denser, with less delimitation. In the reticular dermis, in general, there was an increase in the thickness of the collagen fibers with a decrease in the spaces between them on the 28th day but mainly on the 60th day (Figure 2).

### 3.2. Viability of Bacilli in hOSEC

To evaluate the viability of bacilli in the explants, 16S rRNA was used as a target RNA for RT-PCR, and all culture times until the 60th day demonstrated amplification of this target, with only four fragments showing undetected expression (Table 2).

### 3.3. Maintenance of Viable and Infective Bacilli in hOSEC

After defining the bacillary viability in the explants until D60, we evaluated whether their infectiveness was also maintained by inoculating bacilli recovered from explants (D28 and D60) into the foot pads of athymic mice. The in vivo experiment included 83 animals, and 21.7% of the mice died before the time of analysis (<5 months) of causes unrelated to the experiment. Euthanasia was performed on the remaining mice after five months of inoculation, and molecular analysis was performed in 43 animals and histological analysis in 65.

Among the mice that received suspensions from D28 explants (N = 37), 10 (27.0%) showed positive microscopy for *M. leprae* by ZN, with counts from 1/100 to 470/100 bacilli/field, and 9 (24.3%) positivity by FF in the histology. Among the mice from D60 explants (N = 28), 5 (17.9%) mice showed positivity by ZN and 1 (5.6%) by FF.

By RT-PCR 16S rRNA analysis, only 15 animals inoculated with suspension from D28 explants and 28 from D60 were analyzed; among these, 6 (40.0%) from D28 and 3 (10.7%) from D60 showed positivity.

Interestingly, higher positivity in the mice was found between the animals inoculated with suspension from fragments originating from male skin (Skin 4). All animals showed positivity 5 months after inoculation in at least one of the analyses. In the viability analyses by 16S rRNA, 60% (6) of the animals from the D28 group and 42.9% (3) of the animals from the D60 group showed amplification of 16S rRNA (Table 3, Figure 3).

### 3.4. hOSEC for Studying the Interaction between M. leprae and Human Skin

Considering that we used an ex vivo model of human skin that remains metabolically active over time, in order to show whether *M. leprae* in skin could interfere with tissue immunity, the relative expression of TGF-β, TNF-α, IFN-γ, IL-1β, IL-10, and IL-8 was measured by RT-qPCR in three female skin samples. IFN-γ and IL-1β showed no detectable amplification in any of the explants, with or without bacilli; with respect to the other cytokines, the presence of *M. leprae* notably modulated the expression by skin compared to the control (saline). 

Notably, viable and dead *M. leprae* inoculated in the skin modulated differently the expression of the cytokines, which further reinforces that the two inoculums are different and that the bacillus remained viable during the culture.

In general, viable bacillus inhibits the expression of TGF-β, IL-10, and TNF-α, while, in contrast, skin with dead bacillus has increased expression of these same targets. At the analysis times of D0 and D60, the expression pattern by live bacillus was different, resembling the dead bacillus.

For IL-8, the opposite pattern of expression between viable and dead remains, but inversely, as viable *M. leprae* stimulated expression and dead inhibited it (Figure 4).

## 4. Discussion

Described 150 years ago, the intriguing *M. leprae* has challenged and hampered leprosy research. Currently, there is no axenic medium capable of cultivating this mycobacterium and the animal models are limited for assessment of the clinical aspects of the disease [22,23,24]. In the present study, we showed a new ex vivo model for maintaining viable *M. leprae* and, additionally, for studying the interaction between bacilli and human skin.

Human organotypic skin explant culture (hOSEC) is an ex vivo model of human skin that, in addition to containing keratinocytes and fibroblasts, maintains the complexity of skin composed of other cell types (melanocytes, Langerhans cells), extracellular matrix (glycosaminoglycans, collagen), and skin structures, such as nerve filaments, vessels, and glands. Ex vivo skin is already used for studies on healing, testing cosmetics, and drug absorption [25]. In culture, some authors have shown that the skin maintains its natural architecture for 14 days [26,27]. Because of the long period of multiplication of *M. leprae* (approximately 14 days), we cultured skin for 60 days in the present study. Frade and collaborators [13] performed studies of a similar period and showed the maintenance of dermal junctions for 75 days in culture and the presence of cells in the basal layer stained for Ki-67, a marker of nuclear proliferation expressed in the cell cycle phases G1, S, G2, and M but not in G0.

We performed qualitative analysis of histological sections, in which it was possible see the histomorphology of the four skins used, and these skins maintained their natural architecture until the 14th day and their viability until the 60th day, exhibiting intact dermo–epidermal junctions in both explants, regardless of the presence of *M. leprae*.

In the natural process of maintenance of the skin, keratinocytes from the basal epidermal layer undergo differentiation after division and migrate to the horny layer during the maturation process, during which their interior is filled with keratins and their nucleus is hydrolytically degraded [28,29]. In the present study, by histology (Figure 2), we observed the keratinocytes maturation process and stratum corneum thickening. The finding of stratum corneum thickening and the reduction in keratinocyte layers indicate the maturation of the cells, although without rapid keratinocyte replacement. Xu and collaborators [30] previously showed that cultured skin exhibits an approximate 20% decrease in the rate of proliferation of basal layer keratinocyte, leading to a reduced ability to maintain the thickness of the epidermis.

The most common natural habitat of *M. leprae* is the human skin, where it survives and multiplies mainly inside macrophages and Schwann cells, in a process that involves the axons of the peripheral nerve system, and it has the ability to parasitize other types of cells that make up the skin [31,32]. Thus, it seemed to us promising to challenge and to evaluate the survival and proliferation of *M. leprae* in an ex vivo model of human skin, considering that this model maintains the histomorphological characteristics of host skin, including epithelial cells, fibroblasts, Langerhans cells, glands, and nerves, in addition to maintaining a limited immune response against the pathogen compared to the complete immunity in the host.

Assays for determining bacillary viability besides the subjective morphological index include the use of fluorescent dyes as markers of membrane integrity [10,33]; the evaluation of cellular biochemical metabolism using radiorespirometry [34]; and the measurement of protein synthesis by molecular biology techniques. Due to its short half-life, RNA has been successfully used as an indicator of viability for several pathogens [35,36,37].

Molecular analysis is more sensitive and specific and is more reliable for evaluating cellular viability. Martinez and collaborators [20] demonstrated a good correlation between 16S rRNA RT-PCR and clinical disease and better sensitivity of 16S rRNA than sodA targets for monitoring leprosy therapy. More recently, Collins and collaborators [38] mention that 16S rRNA is a stable marker that could still be detected in dead bacilli. However, 16S rRNA has been used by other researchers to evaluate the viability of *M. leprae* in clinical, environmental, and laboratory samples [14,39,40,41,42,43,44]. Here, we used 16S rRNA RT-PCR to show molecular viability based on the presence of rRNA until the 60th day of the hOSEC incubation period; furthermore, to corroborate this finding, we demonstrated that *M. leprae*, cultivated in hOSEC, beyond maintaining viability, also had its infectivity preserved because it was able to infect the foot pad of several athymic mice.

Several attempts to cultivate *M. leprae* in an axenic medium have been unsuccessful or unreproducible and have resulted in the loss of the ability of the bacilli to infect animals, as described in the 1930s, Lima (1937) [9] in the tentative to reproduce the protocol used by Vaudremer (1935), reported that the long-term maintenance of M. leprae obtained from a leprosy patient in a culture medium formulation failed, as the bacilli lost their pathogenic characteristics and became non-viable. Other groups, more recently, tried to use a medium under microaerophilic conditions [8,45], but within weeks, the bacilli gradually lost their capacity to grow in artificial media and survived for no more than 36 weeks of incubation.

Here, we showed the maintenance of viable bacilli, well-stained bacilli, and detectable levels of 16S rRNA in explants. In addition, it has been demonstrated that after 60 days in explants these bacilli were able to infect mice and remain viable for five months in mouse foot pads, as demonstrated by ZN/FF staining and RT-PCR of macerated foot pads. Unfortunately, the number of animals analyzed was affected by deaths before the analysis time and by an error in handling the paw after euthanasia, mainly in the D60 group, but the results, mostly from Skin 4, are encouraging, showing the maintenance of infectious bacilli after culture in the hOSEC.

Amako and collaborators [7] cultivated *M. leprae* bacilli in modified Kirchner medium containing several nutrients (egg yolk extract, pyruvate, and transferrin) and human plasma, and maintained the bacilli for over 120 days, but without observing signs of exponential growth and not all cells that constituted the colony as globi divided or survived: *M. leprae* seems to have unusual replication cycles. Ferreira and collaborators [39] showed improved conditions when *M. leprae* was grown in arthropod cells. In hOSEC, our results showed viability and infectivity of bacilli for 60 days; however, others experiments will be necessary to address the rate of multiplication of the bacilli in this model.

Some recent studies that demonstrated successfully maintenance of viable and replicating bacilli in protozoa, enriched medium, and arthropod cells [7,39,46], using large amounts of bacilli, on the order of 10^7^, and incubation at low temperatures, approximately 30 °C to 32 °C, considered ideal temperatures for *M. leprae* growth [47,48]. In our experiment, we have considered 37 °C for skin maintenance in culture, but low temperature should be considered in future experiments with *M. leprae*.

In addition to maintain viable bacilli in hOSEC, this model proved that it can be used as one useful tool to assess the interaction between *M. leprae* and human skin because it has been demonstrated that the bacilli modulated gene expression in the skin. The cytokines TGF-β, TNF-α, IL-8, and IL-10 have important roles in the inflammatory process, responses to pathogens, and in other processes, such as the healing process, cell differentiation, and cell migration [49,50,51,52,53,54], that are certainly triggered in explants after the excision of the source tissue, and after interactions with the bacilli.

Regarding the processes of cell differentiation and migration, some studies have reported that bacilli interfere with them; for example, macrophages promote mycobacterial spread during early infection [31,55], and in the Schwann cells, in which *M. leprae* shuts off the differentiation program, bacilli change the characteristics of differentiated cells and alter the expression of the genes involved in mesenchymal endothelial transition, promoting their survival and dissemination [3,31]. In this study, clearly, viable bacilli, unlike dead ones, inhibited TGF-β expression, an important molecule involved in, among other processes, cell differentiation and proliferation.

TNF-α and IL-10, molecules involved in the amplification of the inflammatory response, in the control of the Th1/Th2 balance, and which, in leprosy patients, have an important role in the different forms and reactional states [56,57], in hOSEC had their expression rates reduced by viable bacilli and increased by dead bacilli, compared to skin without bacilli. IL-8, with mainly chemotactic action on cell migration for the immune response [58], also had its gene expression differentially modulated between viable and dead bacilli, showing very low rates when the tissue was challenged with dead bacilli, and showing an increase in its expression in the presence of viable bacilli.

This primary insight about differential modulation for viable and dead bacilli in the skin reinforces the important consideration of the performance of dead *M. leprae* during and after multidrug therapy. Additionally, it corroborates the discussion on the different performances of the expression profile induced by viable and dead bacilli, for example, in neuropathy [59], and it is important for understanding the prognosis of patients and the mechanisms that *M. leprae* uses to challenge and subvert the immune system. In addition, this experiment further reinforces that the two inocula (viable and dead *M. leprae*) are different and that the bacillus remained viable during the culture periods.

Our results successfully demonstrated the maintenance of viable *M. leprae* in an ex vivo human skin model for up to 60 days, while maintaining their infective potential, demonstrated by results from athymic mice, and showed the influence of *M. leprae* in modifying the immunological skin response. These important and unpublished findings support the development of further experimental models for studies of *M. leprae* biology and its interactions, as well as clinical, immunological, and drug susceptibility mechanisms.

This is the first time that human skin remains have been used as a model for maintaining *M. leprae* in the laboratory, and it has been proved to be possible. This first work opens up prospects for further studies and improvements to the hOSEC model itself, as well as helping to reduce the number of experimental animals used, especially for leprosy research.

## Figures and Tables

**Figure 1 tropicalmed-09-00135-f001:**
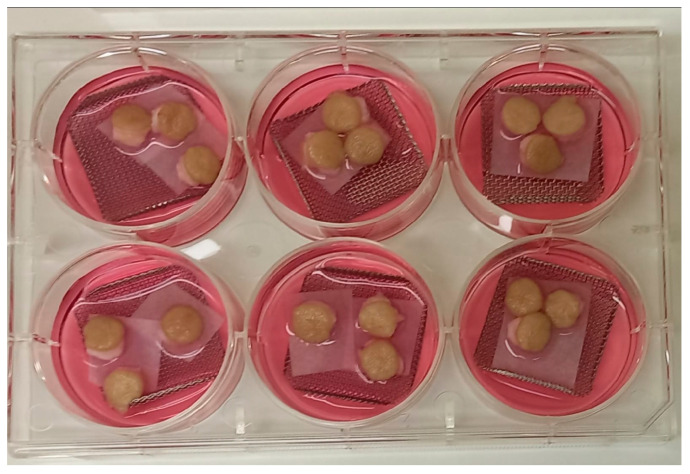
Photo of hOSEC explants on grids in culture plates. Explants of skin obtained from 0.8 cm diameter punch. The explants were placed on filter paper and metal grids in a six-well culture plate with DMEM.

**Figure 2 tropicalmed-09-00135-f002:**
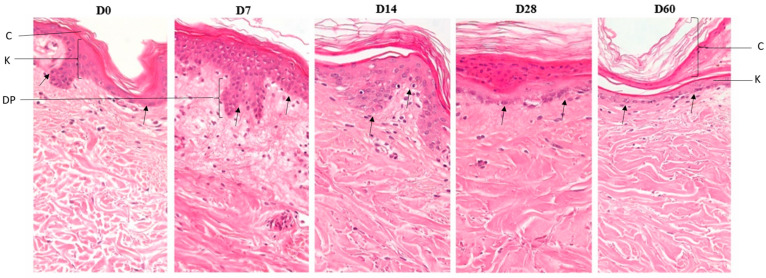
Histomorphology of skin from hOSEC. Sections of skin explants stained with hematoxylin and eosin, from left to right, with culture times of D0, D7, D14, D28, and D60. The images were obtained with a 40× objective and Leica Application Suite (LAS) software version 3.2.0 using the Stretch Image tool. C: corneal layer; K: epidermis, keratinocyte layer, DP: dermal papilla; arrow: basal layer, dermo–epidermal junction. Notably, no inflammatory processes were observed with the recruitment of defense cells.

**Figure 3 tropicalmed-09-00135-f003:**
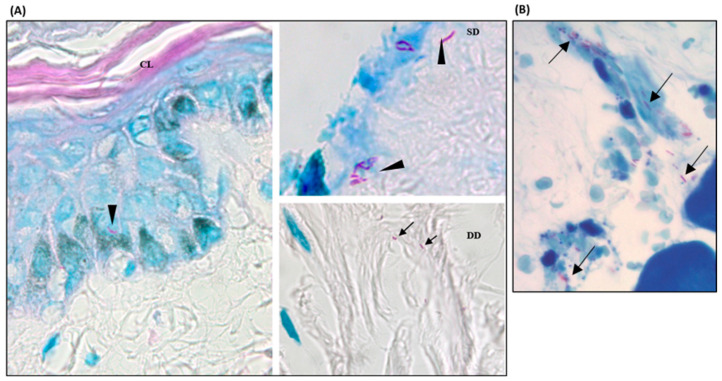
*M. leprae* in skin from hOSEC and in foot pads from athymic mice inoculated. Fite-Faraco staining (bacillus-specific stain) showing intact and well-stained bacilli. 1000× magnification. (**A**) Histological section from explant with *M. leprae* showing bacilli in the keratinocyte layer (arrowhead) and superficial and deep dermis regions (arrow). CL = corneal layer, SD = superficial dermis, DD = deep dermis. (**B**) Histological section from athymic mouse foot pad after 5 months of inoculation with a suspension from D28 culture, the arrow showing bacilli.

**Figure 4 tropicalmed-09-00135-f004:**
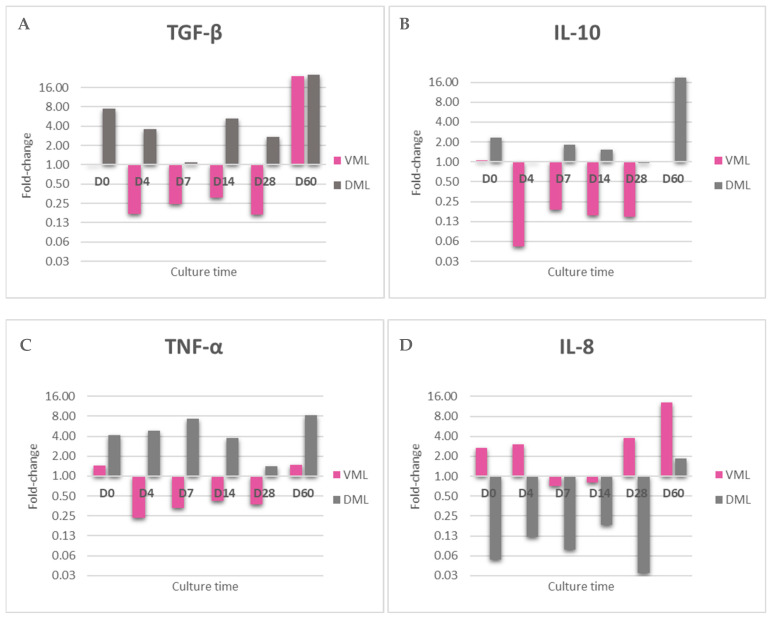
Interaction between M. leprae and skin. Relative gene expression of cytokines (**A**) TGF-β, (**B**) IL-10, (**C**) TNF-α and (**D**) IL-8 in fragment with viable M. leprae (VML) and dead M. leprae (DML) comparing with saline group during the follow-up period, calculated by the 2^ΔΔCT^ method based on CT value obtained from RT-PCR using 18S rRNA as reference gene and the Saline group as Normalizer. The analysis was performed on 3 female skin.

**Table 1 tropicalmed-09-00135-t001:** Primer sequences and cycling protocol.

Target	Sequence	Cycling	Reference
*16S rRNA*	5′ TCGAACGGAAAGGTCTCTAAAAAATC 3′ 5′ CCTGCACCGCAAAAAGCTTTCC 3′	2 min at 95 °C; 45 cycles of 2 min at 94 °C, 2 min at 60 °C, and 3 min at 72 °C; and 10 min at 72 °C	[14]
*18S rRNA*	5′ GTAACCCGTTGAACCCCATT 3′ 5′ CCATCCAATCGGTAGTAGCG 3′	2 min at 95 °C; 38 cycles of 94 °C for 2 min, 60 °C for 2 min, and 72 °C for 3 min; and 72 °C for 10 min.	[15]
*IL-1β*	5′ CTTCATCTTTGAAGAAGAACCTATCTTCTT 3′ 5′ AATTTTTGGGATCTACACTCTCCAGCTGTA 3′	95 °C for 2 min; 45 cycles of 95 °C for 5 s, 62 °C for 10 s, and 72 °C for 20 s.	[16]
*TGF-β*	5′ ACATCAACGCAGGGTTCACT 3′ 5′ GAAGTTGGCATGGTAGCCC 3′	95 °C for 2 min; 45 cycles of 95 °C for 5 s, 60 °C for 10 s, and 72 °C for 20 s.	[17]
*TNF-α*	5′ TGGCTTTCACATACTGCTGGTA 3′ 5′ GCTGGTTATCTCTCAG CTCCA 3′	95 °C for 2 min; 45 cycles of 95 °C for 5 s, 60 °C for 10 s, and 72 °C for 20 s.	[17]
*IFN-γ*	5′ GGCTTTTCAGCTCTGCATCG 3′ 5′ TCTGTCACTCTCCTCTTTCCA 3′	95 °C for 2 min; 45 cycles of 95 °C for 5 s, 60 °C for 10 s, and 72 °C for 20 s.	[17]
*IL-8*	5′ ACCGGAAGGAACCATCTCAC 3′ 5′ AAACTGCACCTTCACACAGAG 3′	95 °C for 2 min; 45 cycles of 95 °C for 5 s, 60 °C for 10 s, and 72 °C for 20 s.	[17]
*IL-10*	5′ TGAGAACCAAGACCCAGACA 3′ 5′ TCATGGCTTTGTAGATGCCT 3′	95 °C for 2 min; 45 cycles of 95 °C for 5 s, 60 °C for 10 s, and 72 °C for 20 s.	[17]

**Table 2 tropicalmed-09-00135-t002:** Cycle threshold of 16S rRNA by RT-PCR.

Day	Skin 1	Skin 2	Skin 3	Skin 4
D0	28.05	34.73	30.29	UR
	37.32	**38.30**	33.51	UR
	ND	35.34	34.95	ND
D4	28.66	32.66	28.74	29.21
	28.45	31.70	29.84	28.17
	**27.70**	31.08	**28.01**	**28.03**
D7	29.21	30.75	33.35	28.62
	29.77	30.27	33.19	29.16
	29.70	33.21	29.43	31.28
D14	29.71	29.80	29.80	29.12
	29.01	**29.19**	28.92	30.10
	30.00	30.54	28.79	31.37
D28	31.03	30.65	28.37	**33.13**
	31.27	32.37	31.61	29.31
	31.43	33.82	**33.70**	ND
D60	31.30	UR	29.81	30.61
	37.17	UR	30.22	ND
	**39.71**	UR	32.90	UR

CT value of 16S rRNA RT-PCR in each explant inoculated with bacilli during the follow-up period. The lowest and highest CT values for each skin sample are in bold. ND = undetected; UR = not performed.

**Table 3 tropicalmed-09-00135-t003:** Positivity in foot pads inoculated with bacilli from hOSEC after 28 and 60 days of culturing.

Inoculum (nº of Animals)	ZN (+) (%)	FF (+) (%)	RT-PCR (+) (%)	At Least One Microscopy Analysis (+) (%)	ZN and RT-PCR (+) (%)	FF and RT-PCR (+) (%)	At Least One Microscopy Analysis (+) and RT-PCR (+) (%)
D28 (10)	6 (60.0)	5 (50.0)	6 (60.0)	8 (80)	4 (40.0)	4 (40.0)	6 (60.0)
D60 (7)	3 (42.9)	0 (0.0)	3 (42.9)	3 (42.9)	1 (14.3)	0 (0.0)	1 (14.3)
Total (17)	9 (52.9)	5 (50.0)	9 (52.9)	11 (64.7)	5 (29.4)	4 (40.0)	7 (41.2)

Positivity (+) for *M. leprae* in the foot pads of mice after 5 months of inoculation with bacilli suspension from male explants (Skin 4). D28 = inoculum from explants in culture for 28 days; D60 = inoculum from explants in culture for 60 days. ZN = Ziehl–Neelsen, performed on one macerated foot pad; FF = Fite-Faraco, performed on half of the second foot pad; RT-PCR= RT-PCR 16Sr RNA performed on the other half of the second foot pad.

## Data Availability

The original contributions presented in the study are included in the article, further inquiries can be directed to the corresponding author.

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
