# Peer review of "Human Skin as an Ex Vivo Model for Maintaining Mycobacterium leprae and Leprosy Studies"

_tropicalmed, 2024, doi:10.3390/tropicalmed9060135_

Round 1

Reviewer 1 Report

Comments and Suggestions for Authors

Dear Authors,

I read your article with interest, since one of my friends in the past has done some work on this too. I read it as a clinician with some experimental and immunological knowledge.

I have a few remarks:

Introduction

Line 40-41: 80% of all humans will genetically never develop leprosy. Only in the remaining 20% 

                         M.leprae subverts the immune system. You are thus talking here about these 20%.

Material and methods

Line 92. Since only 20% of individuals allows M. leprae to multiplicate in the tissues. Were these individuals related to leprosy patients?

I am impressed by the precise description of the methods.

But all was done with Thai-53 strain a “tame” strain. I hope you will repeat this experiment with a wild strain.

Line 197: I have no problem with the statistics.

Results:

Was taken in account that M. leprae is able to survive under suitable conditions for quite some time.  And was there any sign of multiplication in the transplants?

Could the male patient have been belonging to  the 20% who are able to develop leprosy as a disease?

Line 309: Why on D0 too? On D60 I understand many more dead bacilli. Or were there on D0 so may just death and being freshly inoculated their result superseded the effect of the live bacilli?

Discussion:

Line 351: M. leprae can survive under many conditions. But indeed, there is a possibility to study the interaction with the skin.

Line 359: Important is the observation that only for 2 weeks the skin keeps its normal structure. What does that mean about the function after 2 weeks?  I do not know. I am not convinced by your observations on normality. This does say nothing about a “normal“ reaction of the cells concerning production of cytokines when exposed to M.leprae. (But I agree it may stay the same).

Line 386; here you mention a limited immune response.  Have you any idea what change occurs?

Line 412: It seems better than other experimental conditions.

Line 427-433: Indeed, now show multiplication. You need skin of the 20% who could maintain M.leprae. So, not just temperature.

Line 438-440: Important observation.

The difference between live and death bacteria oa on the immunology is well known but nicely shown by you. Indicating some use of your model studying different antigenic expressions.

Line 472: I think you have not proven multiplication. But only survival.

Author Response

Dear,

We are very grateful for the evaluations and comments that have greatly improved our work. After reviewing the technical writing and the English language, we decided to make changes to the title, making it more synthetic and objective in relation to what we did and other small corrections to the text that are marked in red letters.

Additionally, we respond to all doubts raised by respected reviewers and our modifications are also described in red text.

We remain at your disposal for any future corrections.

New title: “Human skin as an ex vivo model for maintaining Mycobacterium leprae and leprosy studies”

Reviewer 2 Report

Comments and Suggestions for Authors

I have reviewed the article: "Human skin as an ex vivo model for Mycobacterium leprae maintenance and studies of leprosy."

The title properly reflects the subject of the paper. The study design is appropriate, and the findings are sound and interesting. The authors have systematically presented their findings and have made their case correctly. The figures and tables are appropriate and informative. The discussion part ties together the results of their study.

The language can be improved. There are many grammatical errors noted.

Nine-banded armadillos have been used in leprosy research, but this article does not discuss their use. Also, how does this human skin model compare with the existing animal models?

Thank you.

Comments on the Quality of English Language

Moderate improvement in the English language is required.

Author Response

(The authors gave the same response as above.)
